# Enhanced Tensile Properties of Multi-Walled Carbon Nanotubes Filled Polyamide 6 Composites Based on Interface Modification and Reactive Extrusion

**DOI:** 10.3390/polym12050997

**Published:** 2020-04-25

**Authors:** Min Park, Ji-un Jang, Jong Hyuk Park, Jaesang Yu, Seong Yun Kim

**Affiliations:** 1Korea Institute of Science and Technology (KIST), Seoul 02792, Korea; minpark@kist.re.kr (M.P.); jju204@naver.com (J.-u.J.); hyuk0326@kist.re.kr (J.H.P.); 2KHU-KIST Department of Converging Science and Technology, Kyung Hee University, Seoul 02447, Korea; 3Division of Polymer-Nano & Textile Engineering, Chonbuk National University, Jeonju-si 54896, Korea

**Keywords:** polymer-matrix composites, carbon nanotubes, micro-mechanics

## Abstract

To induce uniform dispersion and excellent interfacial properties, we adopted a strategy of combining both polyamide 6 (PA6) grafting for multi-walled carbon nanotubes (MWCNTs) and reactive extrusion of PA6 matrix, based on anionic ring-opening polymerization of ε-caprolactam (CL). Compared to –COOH and –NCO treatments of MWCNTs, enhanced MWCNT dispersion and tensile properties of the composites were achieved using the applied strategy, and the tensile strength and modulus of the PA6-grafted MWCNT-filled PA6 composites were 5.3% and 20.5% higher than those of the purified MWCNT-filled PA6 composites, respectively. In addition, they were almost similar to the theoretical ones calculated by the modified Mori–Tanaka method (MTM) assuming a perfect interface, indicating that the tensile properties of MWCNT-filled PA6 composites can be optimized by PA6 grafting and reactive extrusion based on the anionic ring-opening polymerization of CL due to uniform MWCNT dispersion and excellent interfacial property.

## 1. Introduction

Since their discovery [1], carbon nanotubes (CNTs) have been of great interest as an efficient reinforcement that can improve the mechanical properties of polymer composites due to their excellent mechanical properties such as 11–150 GPa tensile strength and 0.4–4.15 TPa tensile modulus [2,3,4]. However, there have been few reports on the successful development of CNT-based high strength polymer composites to meet researchers’ expectations [5,6]. Uneven dispersion or aggregation of CNTs in a composite material due to van der Waals force has been pointed out as one of the obstacles to the development of high strength composite material using CNTs [4,7,8]. In addition, it is difficult to induce interfacial bonding between nano-sized CNTs and polymers based on a mechanical interlocking mechanism, so they exhibit limited interfacial properties [4,9,10]. In order to overcome these problems, proper CNT surface treatment is required in consideration of the characteristics of the polymer matrix and the manufacturing process of composite materials.

Appropriate CNT surface treatment not only improves the dispersion of CNTs in composites but also maximizes the interfacial properties between CNTs and polymeric matrices via mechanisms such as hydrogen bonding, covalent and noncovalent bonding, and polymer interdiffusion [11,12,13]. CNTs can be treated with an acid to introduce –COOH groups, which can induce a hydrogen bond or a covalent bond with the polyamide 6 (PA6) matrix [14,15]. The interfacial bonding between CNTs and PA6 matrix can also be improved by introducing –NCO groups on the surfaces of CNTs [16,17]. A previous study [18] reported that PA6 polymers can be grafted on the surface of CNTs, and PA6 polymers on the surface of CNTs are interdiffused with PA6 matrix, resulting in excellent dispersion as well as improved interfacial properties [7,16,19,20].

The manufacturing process of composite materials filled with CNTs can be classified into three main types based on the dispersion method: melt mixing, solution mixing, and in-situ polymerization [4,6,21,22,23]. The melt mixing method is the most economical but is difficult to induce sufficient dispersion. Solution mixing makes it easy to generate uniform dispersion, but it is not economical due to the use of a solvent and the long drying process. For this reason, an in-situ polymerization process based on an economical melting process can be selected, while inducing sufficient dispersion. In particular, it has been widely reported that the reactive polymerization of thermoplastic matrix can induce excellent dispersion of nanofillers, thereby substantially improving the bulk properties of the composites filled with the well-dispersed nanofillers [18,19,20,21,22,23,24,25,26,27,28,29,30].

Although reactive extrusion using anionic polymerization of ε-caprolactam (CL) is an appealing composite process for nanofiller dispersion in terms of both the cost of raw materials and the characteristics of matrix materials, there have been few systematic studies simultaneously applying both reactive extrusion based on the anionic polymerization of CL and the various surface treatments of multi-walled carbon nanotubes (MWCNTs). In this study, –COOH, –NCO, and –PA6 functionalized MWCNTs were prepared, and PA6 composites filled with the surface treated MWCNTs were fabricated by the reactive extrusion based on the anionic polymerization of CL. MWCNT dispersion and tensile properties of the composites were evaluated in order to investigate the effects of the MWCNT surface treatments and the reactive extrusion. By comparing between the measured tensile properties and the theoretically calculated ones based on the modified Mori–Tanaka method (MTM), interfacial properties of the composites were evaluated.

## 2. Experimental

### 2.1. Materials

MWCNTs supplied from the Jeio Co. (Inchon, Korea) were used as a reinforcement. Pristine MWCNTs were purified by sonication in 3 M HCl solution for 6 h and then filtered using a polytetrafluoroethylene (PTFE) filter. The purified MWCNTs were washed with distilled water until reaching pH 7, then vacuum-dried at 80 °C. CL supplied by Capro (Seoul, Korea) was used as a matrix. Sodium hydride (NaH, Sigma-Aldrich, Yongin, Korea) and hexamethylene diisocyanate (HDI, Sigma-Aldrich, Yongin, Korea) were used as a catalyst and activator, respectively, for anionic polymerization of CL. The mechanism of anionic polymerization of CL has been widely investigated and reported as shown in Appendix A [16].

### 2.2. Surface Treatment of MWCNT

#### 2.2.1. Carboxylated MWCNT

First, 1 wt% purified MWCNT was added to the mixed solution of HNO_3_/H_2_SO_4_ at a ratio of 1:3. The mixture was sonicated at 80 °C for 6 h and then filtered using a PTFE filter. The prepared MWCNT was rinsed with distilled water until reaching pH 7, then vacuum-dried at 80 °C to prepare carboxylated MWCNT.

#### 2.2.2. Isocyanated MWCNT

First, 1 wt% carboxylated MWCNT was sonicated in anhydrous acetone, and then 4 vol% HDI was added. The reaction was induced by sonication under dry N_2_ gas at 50 °C for 24 h, then filtered using a PTFE filter.

#### 2.2.3. PA6 Grafted MWCNT

Isocyanated MWCNTs were mixed with molten CL, sonicated at 110 °C for 6 h, and then mixed with molten CL containing NaH catalyst. The CL was propagated from the MWCNT surface, resulting in a PA6-grafted MWCNT in which the MWCNT and PA6 were covalently linked.

### 2.3. Composite Fabrication

The NaH was added to the CL melt, which was removed to a water content of 50 ppm or less. The reaction proceeded at 110 °C until the hydrogen gas generation was completed, and then the air was sealed. Then, the HDI was added to the CL melt, which was removed to a water content of 50 ppm or less. After reacting at 110 °C for 1 h, it was sealed with the air removed. The prepared mixture was reactive extruded using a twin-screw extruder (L40/D19, Bautek CO, Uijungbu, Korea) under optimized process conditions that were derived as shown in Appendix A. At this time, the residence time of the mixture in the extruder was 100 s. To determine the optimum reaction temperature and concentration which would set the reaction completion time to be shorter than the residence time of the mixture, anionic polymerization of the CL proceeded under various conditions. To investigate the effect of MWCNT surface treatment on the structural and tensile characteristics of composites, 1 wt% purified, carboxylated, isocyanated, and PA6-grafted MWCNTs were dispersed in the activator mixture and then mixed with the catalyst mixture. In the reactive extrusion process, PA6 composites were prepared by the anionic polymerization of the CL mixture matrix as shown in Figure 1.

### 2.4. Characterization

Fourier transform infrared spectroscopy (FTIR, Nicolet 6700, Thermo Scientific, Waltham MA, USA) analysis was performed to observe the chemical changes of MWCNTs by the surface treatments. The FTIR spectrum was measured at 8 cm^−1^ resolution in the wave number range of 500–4000 cm^−1^. To observe the compatibility between the prepared MWCNTs and PA6 matrix, the precipitation of MWCNTs in the formic acid, a good solvent of PA6, was observed for 20 days after sonication. The structural changes of MWCNTs were observed by the surface treatments using transmission electron microscopy (TEM, Tecnai F20, FEI Corp., Hillsboro OR, USA). In order to prepare the TEM samples, the purified, carboxylated, isocyanated, and PA6-grafted MWCNTs were dispersed in ethanol and sonicated for 5 min, then the dispersion was dropped on the lacey-carbon coated film on the Cu grid. TEM observations were performed at 120 kV, and the exposure time of accelerating electrons was minimized in an attempt to avoid electron exposure damage during TEM analysis. The molecular weights of the PA6 and composite samples were determined using gel permeation chromatography (GPC, EcoSEC, Tosoh Bioscience, Tokyo, Japan). Samples dissolved in hexafluoroisopropanol with trifluoroacetic acid Na salt were left for one day and filtered with a 0.45 μm PTFE filter. The GPC system was calibrated using polymethyl methacrylate standard. To evaluate the MWCNT dispersion, the fractured surface of the prepared composite was observed using a field emission scanning electron microscope (FE-SEM, Nova NanoSEM 450, FEI Corp., Hillsboro OR, USA) at a voltage of 15 kV after coating (Pt) under vacuum for 80 s by a sputter machine (Ion Sputter E-1030, Hitachi High Technologies, Tokyo, Japan). The tensile test was carried out using a universal tester (Instron 5567, Norwood, MA, USA) with a crosshead speed of 5 mm/min at room temperature according to American society for testing and materials (ASTM) D638.

## 3. Micromechanics Modeling Approach

The Mori–Tanaka micromechanical approach has mainly been used to estimate effective heterogeneous material properties, particularly for composites containing low volume fractions of reinforcements in elastic resins. Yu et al. [31] developed a modified MTM for predicting effective elastic properties for nanocomposites containing multiple nano heterogeneities with arbitrary shapes and orientations.

### 3.1. Modified Mori-Tanaka Method for Composite Stiffness Properties

This approach may be extended to the case for composites containing multiple distinct heterogeneities. Suppose that the matrix contains m distinct types of ellipsoidal heterogeneities (*k* = 1, 2,..., *m*), each consisting of *n_k_* layers (*α_k_* = 1, 2,..., *n_k_*; k = 1, 2,..., *m*). Each type of heterogeneity has distinct elastic properties, shape, and orientation distribution. Based on a composite containing m distinct types of heterogeneities (*k* = 1, 2,..., *m*), each with an arbitrary number of layers (*n_k_*) in a matrix (0), the overall elasticity tensor, $\bar{L}$, may be expressed as
(1)$$\begin{array}{c} \bar{L}=L^{\left(0\right)}:\{I+\sum_{k=1}^{m}\left[\sum_{\alpha_{k}=1}^{n_{k}}c_{{}^{\left(k\right)}}{}_{\alpha_{k}}(S_{\left(k\right)}-I):{\left(A_{\left(k\right)}{}^{\left(\alpha_{k}\right)}-S_{\left(k\right)}\right)}^{-1}]\;\right\}:\;\;\;\;\;\;\;\;\;\;\;\\ \;\{I+\sum_{k=1}^{m}{\left[\sum_{\alpha_{k}=1}^{n_{k}}c_{{}^{\left(k\right)}}{}_{\alpha_{k}}S_{\left(k\right)}:{\left(A_{\left(k\right)}{}^{\left(\alpha_{k}\right)}-S_{\left(k\right)}\right)}^{-1}]\;\right\}}^{-1} \end{array}$$
where
(2)$$A_{\left(k\right)}{}^{\left(\alpha_{k}\right)}={\left(L^{\left(0\right)}-L_{\left(k\right)}{}^{\left(\alpha_{k}\right)}\right)}^{-1}:L^{\left(0\right)}$$

$A_{\left(k\right)}{}^{\left(\alpha_{k}\right)}$ is the local strain concentration tensor for the *α_k_*
^th^ layer of the *k*^th^ heterogeneity (*α_k_* = 1, 2,..., *n_k_*, *k* = 1, 2,..., *m*). Here, $L_{\left(k\right)}{}^{\left(\alpha_{k}\right)}$ is the fourth rank elastic stiffness tensor for the *α_k_*
^th^ layer of the *k*^th^ heterogeneity, $c_{{}^{\left(k\right)}}{}_{\alpha_{k}}$ is the volume fraction of the *α_k_*^th^ layer of the *k*^th^ heterogeneity, and ***S***_(*k*)_ is the 4^th^ rank Eshelby tensor common to the heterogeneity and all layers of the *k*^th^ heterogeneity.

### 3.2. Modified Mori-Tanaka Method for Composite Strength Properties

The ultimate applied stresses in heterogeneities and the matrix, ($\sigma_{A}{}^{M}$,$\sigma_{A}{}^{\Omega_{\alpha }}$), can be determined by the average stress reaching the strength properties of the matrix or the heterogeneities [32]. The composite strength property is dominated by the heterogeneity strength value when the ultimate applied stress in heterogeneities ($\sigma_{A}{}^{\Omega_{\alpha }}$) is less than that of the matrix ($\sigma_{A}{}^{M}$). By contrast, the composite strength property is dominated by the matrix strength value when the ultimate applied stress in matrix ($\sigma_{A}{}^{M}$) is less than that of heterogeneities ($\sigma_{A}{}^{\Omega_{\alpha }}$). The ultimate applied stress ($\sigma_{A}{}^{M}$) can be determined by Equation (3) when the average stress in the matrix, ($\sigma_{A}+{\langle \sigma \rangle }_{M}$), reaches the strength value of matrix ($\sigma_{s}{}^{M}$); it can be expressed as follows:(3)$$\sigma_{A}{}^{M}={\left[I-\sum_{\alpha =1}^{n}c_{\alpha }L^{\left(0\right)}:\left(\sum_{\alpha =1}^{n}S^{\left(\alpha \right)}-I\right)\;:{\sum_{\alpha =1}^{n}\alpha_{\alpha }}^{-1}:\left(L^{\left(0\right)}-\sum_{\alpha =1}^{n}L^{\left(\alpha \right)}\right):L^{\left(0\right)}{}^{-1}\right]}^{-1}\;:\sigma_{s}{}^{M}$$

Similarly, the ultimate applied stress ($\sigma_{A}{}^{\Omega_{\alpha }}$) can be determined by Equation (4), where the average stress in the $\alpha^{th}$ heterogeneity ($\sigma_{A}+\sum_{\alpha =1}^{n}{\langle \sigma \rangle }_{\Omega_{\alpha }}$) reaches the strength value of $\alpha^{th}$ heterogeneity ($\sigma_{s}{}^{\Omega_{\alpha }}$); it can be expressed as follows:(4)$$\sigma_{A}{}^{\Omega_{\alpha }}={\left[\left(1-\sum_{\alpha =1}^{n}c_{\alpha }\right)L^{\left(0\right)}:\left(\sum_{\alpha =1}^{n}S^{\left(\alpha \right)}-I\right)\;:{\sum_{\alpha =1}^{n}\alpha_{\alpha }}^{-1}:\left(L^{\left(0\right)}-\sum_{\alpha =1}^{n}L^{\left(\alpha \right)}\right):L^{\left(0\right)}{}^{-1}+I\right]}^{-1}\;:\sigma_{s}{}^{\Omega_{\alpha }}$$

## 4. Results and Discussion

### 4.1. Surface Treatment of MWCNTs

FTIR spectra are shown in Figure 2a to observe the surface functionality of MWCNTs by the applied surface treatments. The FTIR spectrum of the carboxylated MWCNTs showed O–H peaks near 3421 and 1453 cm^−1^, as well as peaks at 1654 and 1710 cm^−1^ due to the C=O of carboxyl groups [16,17]. In the spectrum of the isocyanated MWCNT, the characteristic peak of O–H disappeared, and a new peak of 2283 cm^−1^ was found, which corresponds to the asymmetric elongation of the isocyanate group [16]. The peaks at 2900 and 2825 cm^−1^ were due to the C–H stretching of the methyl group of the diisocyanate [16]. The peak at 1245 cm^−1^ was attributed to the C–N stretch of the carbamate group resulting from the esterification reaction of the hydroxyl group and the isocyanate group [16]. As compared with the FTIR spectrum of the isocyanated MWCNT, the disappearance of the peak at 2283 cm^–1^ in the FTIR spectrum of the PA6-grafted MWCNT indicated that the isocyanate group was consumed during the PA6 polymers grafting reaction [16]. In addition, the characteristic absorption peaks of PA6, such as 3297 (NH stretching), 3060, 1637 (amide I band), and 1540 cm^−1^ (amide II band) were observed [16]. Based on the FTIR results, it was confirmed that the carboxyl group was introduced into the MWCNT surface by acid treatment, the isocyanated MWCNT was prepared by the isocyanate treatment, and the activation of the isocyanate group for the anionic polymerization of CL resulted in the PA6-grafted MWCNT.

TEM images are shown in Figure 3 to observe the structural changes in MWCNTs by the surface treatments. Unlike the other MWCNTs, the PA6-grafted MWCNT was coated with about 8 nm of PA6 layer. In the case of the PA6-grafted MWCNT, the surface of MWCNT was covalently bonded to PA6 polymers, so it can be expected to exhibit improved compatibility with the PA6 matrix. Excellent compatibility between the PA6-grafted MWCNTs and PA6 matrix was observed based on the fact that no precipitation was observed only in the PA6-grafted MWCNT sample as shown in Figure 2b.

### 4.2. Reactive Extrusion

To complete the anionic polymerization of CL during the reactive extrusion process as shown in Figure 1, the reaction completion time must be faster than the residence time in the extruder. The rate of anionic polymerization of CL is known to be a function of reaction temperature as well as catalyst and activator concentrations [18,33,34]. To set an activator/catalyst ratio that can complete the anionic polymerization faster than the residence time of 100 s, the optimized reactive extrusion process conditions are presented in Appendix A; Figure 4a shows the solidification time according to the activator/catalyst ratio. It was confirmed that the appropriate reaction completion time was induced when the ratio of activator in the mixture was 0.4 or 0.6 mol% and the activator/catalyst ratio was higher than 1.5. Since the activator and especially the catalyst are more expensive than CL, the optimum composition of 0.4 mol% activator and 0.6 mol% catalyst, which can minimize the use of these materials, was selected. Based on the optimum activator/catalyst composition as well as the reaction kinetics expression of Wittmer and Gerrens [35] as shown in Figure 4b (details in Appendix A), the optimum reaction temperature was determined. The higher the reaction temperature, the shorter the reaction time. However, at temperatures exceeding 240 °C, there is a risk of ignition and thermal deformation due to the rapid reaction, and 240 °C can be considered as the optimum reaction temperature.

In order to investigate the effect of MWCNT incorporation on the anionic polymerization, the molecular weight of the reactive extruded PA6 matrix is shown in Figure 4c according to the MWCNT content. The reactive extruded PA6 without added MWCNT showed a molecular weight of about 42,000 and a yield of 98%. As the MWCNT content increases, the molecular weight and yield decreased because the mobility of the grown polymer chain was reduced due to the spatial restriction of the MWCNT [36]. When the MWCNT was incorporated within 5 wt% in the optimum process conditions, the molecular weight of the PA6 polymer exceeded 22,000, indicating good mechanical properties [37]. In addition, the effect of MWCNT functional groups on the molecular weight was insignificant, and similar molecular weight and yield were observed. Therefore, when the MWCNT was incorporated within 5 wt%, a molecular weight higher than 22,000, which exhibited the good mechanical properties of PA6 matrix, was induced irrespective of the surface treatment of the MWCNT.

### 4.3. MWCNT Dispersion of Composites

Figure 5 shows the FE-SEM images of the fracture surface of the composites to evaluate the MWCNT dispersion in the PA6 composites produced by the reactive extrusion. In the fracture morphology images, the anionic polymerization of CL derived during the reactive extrusion process resulted in a quite uniform dispersion of MWCNTs. However, certain MWCNT agglomerates were found in the fracture surface of the composites containing the purified MWCNTs. In addition, MWCNT dispersion was further improved by applying the surface treatments. In particular, when PA6-grafted MWCNTs were applied, the uniform dispersion was confirmed.

### 4.4. Tensile Properties of Composites

The tensile properties of PA6 composites filled with 1 wt% MWCNTs are shown in Figure 6, and those of the PA6-grafted MWCNTs filled PA6 composites showed the highest results. The tensile strength and modulus of the PA6-grafted MWCNTs filled PA6 composites were increased by 25% and 171%, respectively, as compared to those of neat PA6. Further, they were 5.3% and 20.5% higher than those of the purified MWCNT-filled PA6, respectively. These improved tensile properties can be attributed to the improved dispersion of the PA6-grafted MWCNT in PA6 composites and the enhanced interfacial properties between the PA6-grafted MWCNT and PA6 matrix. In the case of the PA6 composites filled with the PA6-grafted MWCNT, because the surface of the MWCNT is covalently bonded to PA6 polymers, interdiffusion between the PA6 chain on the MWCNT surface and the PA6 matrix could occur, and the dispersion of the MWCNT in the composite was improved. In addition, amide exchange reactions can occur due to the high processing temperature, and the interfacial properties between them can be improved [38]. In the case of carboxylated and isocyanated MWCNT-filled PA6 composites, the tensile properties were confirmed to be superior to those of the purified MWCNT-filled PA6 composite, and this can be attributed to the hydrogen bonding between the functional groups in the MWCNTs and the PA matrix [39].

Theoretically obtained tensile properties of the PA6 composites filled with 1 wt% MWCNTs are shown in Figure 6. The theoretical results were most similar to the actual results using PA6-grafted MWCNTs, due to their excellent dispersion and interfacial properties. Regarding tensile modulus, when the experimental result of the composite incorporating the PA6-grafted MWCNTs was compared with the theoretical results based on the classical Halpin–Tsai model, assuming the random orientation of MWCNTs (see Appendix A) and based on the MTM assuming a perfect interface, the experimental result was 8.6% and 0.4% lower than the theoretical results, respectively. In the case of tensile strength, as compared with the theoretical results based on the rule of mixture (see Appendix A) and based on the MTM assuming a perfect interface, the measured result was 58.4% and 4.0% lower than the theoretical results, respectively. These results indicated that more realistic effective tensile modulus can be derived by considering the MWCNT shape in the MTM, and that it was necessary to consider the MWCNT shape in the theoretical calculation in order to obtain the effective tensile strength corresponding to the actual value. By applying weakened interfacial properties to the MTM, we could simulate the tensile properties of PA6 composites containing 1 wt% MWCNTs according to the surface treatments. The tensile strength and modulus of the PA6 composites incorporating carboxylated and isocyanated MWCNTs can be calculated by applying the 80% level of interface, while those of the composites incorporating PA6-grafted MWCNT could be simulated by assuming a perfect interface, indicating that PA6 grafting can induce better interfacial properties.

## 5. Conclusions

The tensile properties of MWCNT-filled PA6 composites were optimized by carboxylated, isocyanated, and PA6-grafted functionalizations, as well as reactive extrusion based on the anionic polymerization of CL. Based on the FTIR results, it was confirmed that the functionalization strategy was successfully achieved. The optimum process conditions were derived by analyzing the relationship between the reactive extrusion processing parameters and the resultant molecular weight. The composites filled with the PA6-grafted MWCNTs showed the highest tensile properties because of their uniform MWCNT dispersion and excellent interfacial property. By comparing between the measured tensile properties and the theoretically calculated ones based on the modified MTM, the interfacial property between the PA6-grafted MWCNT and PA6 matrix was evaluated to be an almost perfect interface.

## Figures and Tables

**Figure 1 polymers-12-00997-f001:**
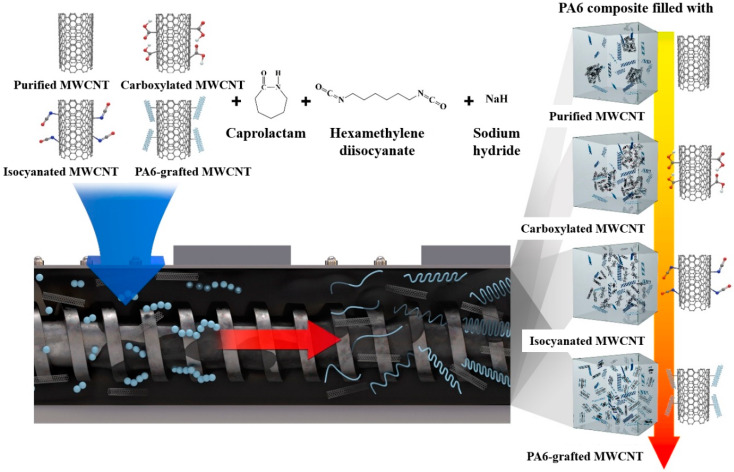
Schematic of the applied strategy.

**Figure 2 polymers-12-00997-f002:**
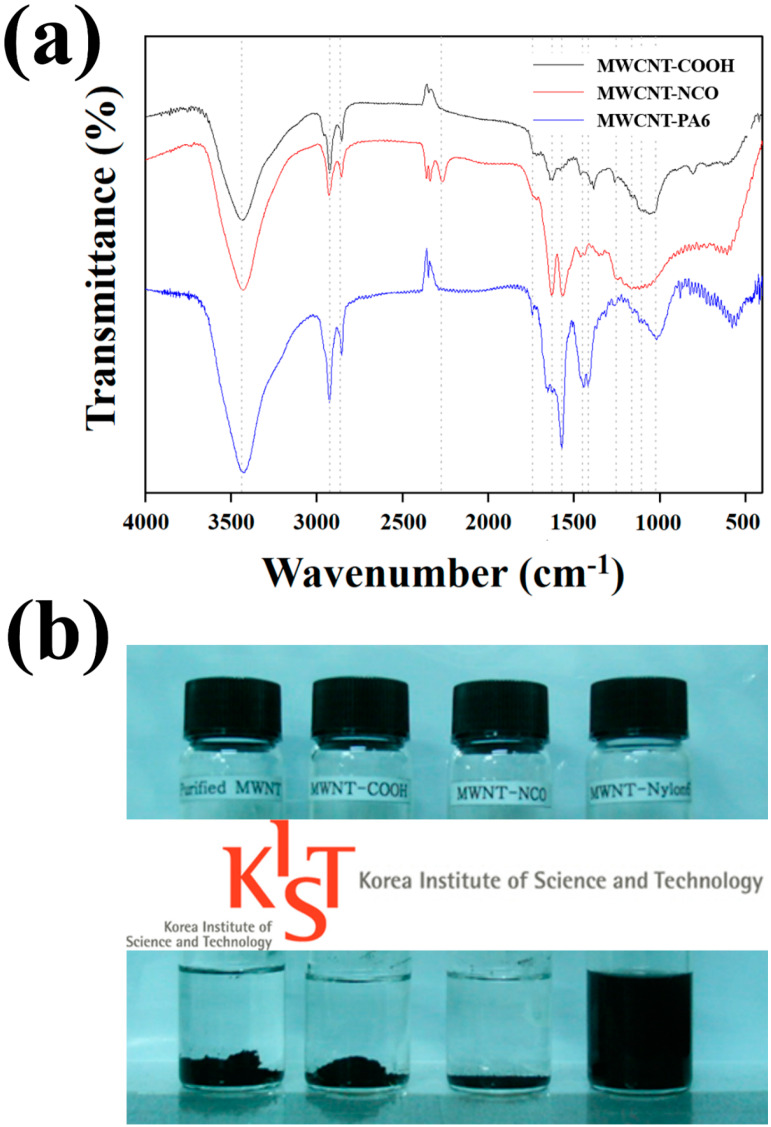
(**a**) Fourier transform infrared spectroscopy (FTIR) spectra of surface-treated multi-walled carbon nanotubes (MWCNTs) and (**b**) MWCNT dispersion in formic acid after 20 days after ultrasonication according to the surface treatment.

**Figure 3 polymers-12-00997-f003:**
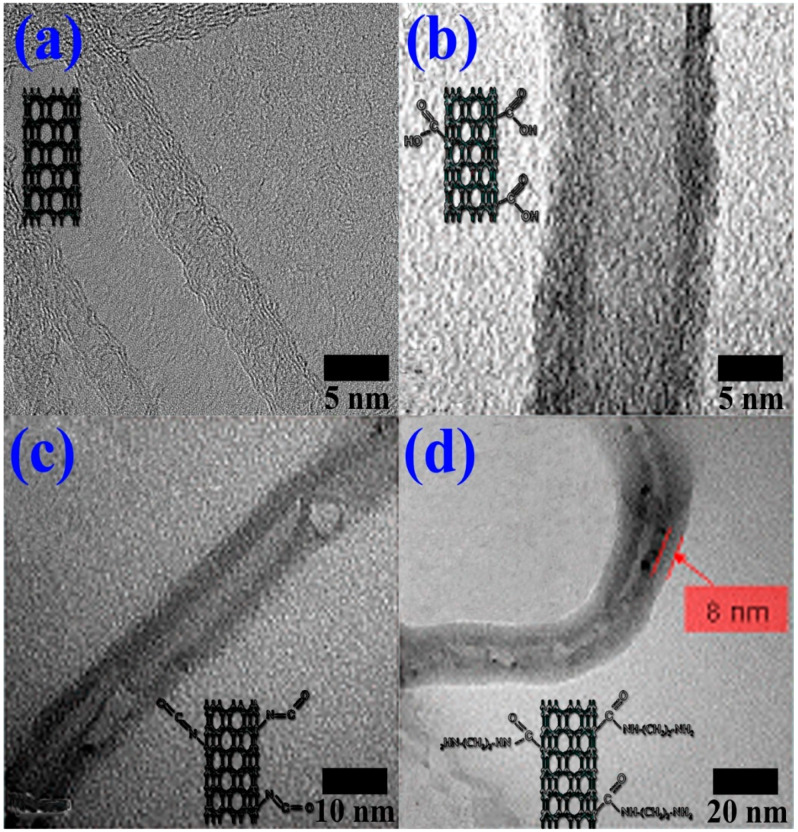
Transmission electron microscopy (TEM) images of (**a**) purified MWCNTs, (**b**) carboxylated MWCNTs, (**c**) isocyanated MWCNTs, and (**d**) polyamide 6 (PA6) grafted MWCNTs.

**Figure 4 polymers-12-00997-f004:**
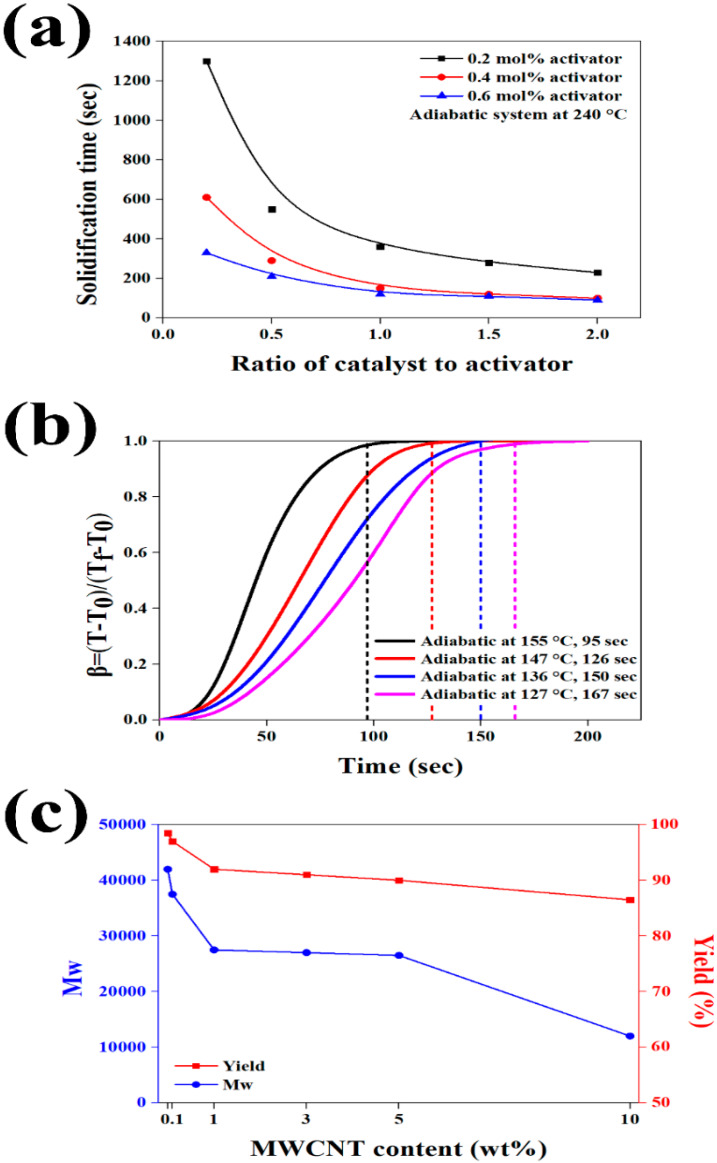
(**a**) Dependence of solidification time on the catalyst to activator ratio at 240 °C, (**b**) reaction kinetics expressions of Wittmer and Gerrens, and (**c**) molecular weight and yield of polymerized PA6 according to MWCNT content.

**Figure 5 polymers-12-00997-f005:**
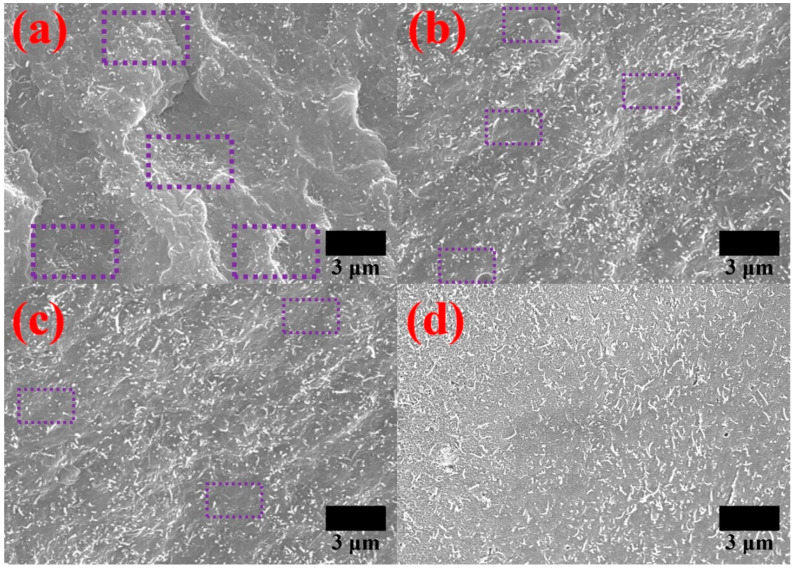
FE-SEM images of the composites filled with (**a**) purified MWCNTs, (**b**) carboxylated MWCNTs, (**c**) isocyanated MWCNTs, and (**d**) PA6-grafted MWCNTs.

**Figure 6 polymers-12-00997-f006:**
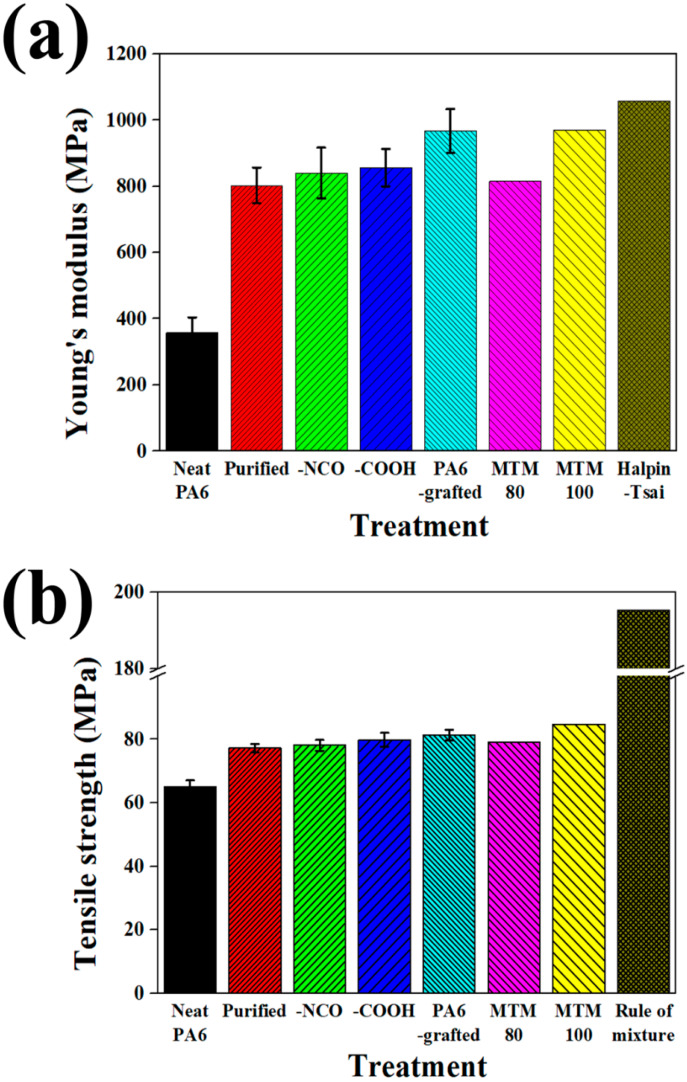
Measured and calculated tensile (**a**) modulus and (**b**) strength of PA6 composites.

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
