# Peer review of "Enhanced Tensile Properties of Multi-Walled Carbon Nanotubes Filled Polyamide 6 Composites Based on Interface Modification and Reactive Extrusion"

_polymers, 2020, doi:10.3390/polym12050997_

Round 1

Reviewer 1 Report

The manuscript is interesting from a technical point of view. It is well written, and conclusions are clearly drawn from results. I therefore suggest publication as is.

Author Response

Response to Reviewer 1 Comments

Point 1: The manuscript is interesting from a technical point of view. It is well written, and conclusions are clearly drawn from results. I therefore suggest publication as is.

Response 1:We would like to thank the reviewer's positive comment.

Reviewer 2 Report

In this manuscript, “Enhanced tensile properties of multi-walled carbon nanotubes filled polyamide 6 composites based on interface modification and reactive extrusion”. This paper adopted a strategy of combining both polyamide 6 (PA6) grafting for multi-walled carbon nanotubes (MWCNTs) and reactive extrusion of PA6 matrix, based on anionic ring-opening polymerization of ε-caprolactam (CL). they were almost similar to the theoretical ones calculated by the modified Mori-Tanaka method (MTM) assuming a perfect interface, indicating that the tensile properties of MWCNT-filled PA6 composites can be optimized by PA6 grafting and reactive extrusion based on the anionic ring-opening polymerization of CL due to uniform MWCNT dispersion and excellent interfacial property.

The article has clear logic, clear thinking, clear reaction mechanism, Summary complete, chart data reasonable, sufficient, necessary,and combined with the formula to explain the problem. But, there are still some shortcomings in the paper. I will give you some my suggestions.

  1. Heading error

In the third section of the article, there are two 3.1 and repeated errors, which should be 3.2.

  1. Scale labeling

Proportion labeling color is not harmonious, or even not clear, it is recommended to use black.

  1. There were some reports for the carbon nanomaterials/polymer composites and interface, such as Chemical Engineering Journal, 2014, 237(0):291-299、Polymer Composites, 2017, 38(1):5-12;Journal of Applied Polymer Science, 2013, 130(2):1194-1202Composites Science and Technology, 2018, 157:57-66;ACS Applied Materials & Interfaces, 2015, 7(21):11583-11591. These need to be reviewed completely.
  2. Horizontal axis of the font is too small (such as 4.4. Tensile properties of composites)

Author Response

Response to Reviewer 2 Comments

In this manuscript, “Enhanced tensile properties of multi-walled carbon nanotubes filled polyamide 6 composites based on interface modification and reactive extrusion”. This paper adopted a strategy of combining both polyamide 6 (PA6) grafting for multi-walled carbon nanotubes (MWCNTs) and reactive extrusion of PA6 matrix, based on anionic ring-opening polymerization of ε-caprolactam (CL). they were almost similar to the theoretical ones calculated by the modified Mori-Tanaka method (MTM) assuming a perfect interface, indicating that the tensile properties of MWCNT-filled PA6 composites can be optimized by PA6 grafting and reactive extrusion based on the anionic ring-opening polymerization of CL due to uniform MWCNT dispersion and excellent interfacial property.

The article has clear logic, clear thinking, clear reaction mechanism, Summary complete, chart data reasonable, sufficient, necessary,and combined with the formula to explain the problem. But, there are still some shortcomings in the paper. I will give you some my suggestions.

We greatly appreciate the positive comments from the Reviewer 2. We have made revisions as detailed below.

Point 1: Heading error

In the third section of the article, there are two 3.1 and repeated errors, which should be 3.2.

Response 1: According to the reviewer’s comment, heading error in section 3.1 was revised to 3.2. (Line 5, Page 5)

Point 2: Scale labeling

Proportion labeling color is not harmonious, or even not clear, it is recommended to use black.

Response 2: According to the reviewer’s comment, the scale labeling colors in Figure 3 and 5 were revised to black.

Point 3: There were some reports for the carbon nanomaterials/polymer composites and interface, such as Chemical Engineering Journal, 2014, 237(0):291-299、Polymer Composites, 2017, 38(1):5-12;Journal of Applied Polymer Science, 2013, 130(2):1194-1202、Composites Science and Technology, 2018, 157:57-66;ACS Applied Materials & Interfaces, 2015, 7(21):11583-11591. These need to be reviewed completely.

Response 3: According to the reviewer’s comment, the references were added to the manuscript. (Line 4, Page 12)

  1. Wang, H.; Li, N.; Xu, Z.; Tian, X.; Mai, W.; Li, J.; Chen, C.; Chen, L.; Fu, H.; Zhang, X. Enhanced sheet-sheet welding and interfacial wettability of 3D graphene networks as radiation protection in gamma-irradiated epoxy composites. Compos. Sci. Technol. 2018, 157, 57-66.

  1. Li, W.; Shi, C.; Shan, M.; Guo, Q.; Xu, Z.; Wang, Z.; Yang, C.; Mai, W.; Niu, J. Influence of silanized low‐dimensional carbon nanofillers on mechanical, thermomechanical, and crystallization behaviors of poly(L‐lactic acid) composites—A comparative study. J. Appl. Polym. Sci. 2013, 130, 1194-1202.

  1. Li, W.; Xu, Z.; Chen, L.; Shan, M.; Tian, X.; Yang, C.; Lv, H.; Qian, X. A facile method to produce graphene oxide-g-poly(L-lactic acid) as an promising reinforcement for PLLA nanocomposites. Chem. Eng. J. 2014, 237, 291-299.

  1. Chen, L.; Li, X.; Wang, L.; Wang, W.; Xu, Z. Effect of the molecular chains grafted on graphene nanosheets on the properties of poly(l‐lactic acid) nanocomposites. Polym. Compos. 2015, 38, 5-12.

  1. Ni, Y.; Chen, L.; Teng, K.; Shi, J.; Qian, X.; Xu, Z.; Tian, Xu.; Hu, C.; Ma, M. Superior mechanical properties of epoxy composites reinforced by 3D interconnected graphene skeleton. ACS Appl. Mater. Interfaces 2015, 7, 11583-11591.

Point 4: Horizontal axis of the font is too small (such as 4.4. Tensile properties of composites)

Response 4: According to the reviewer’s comment, the font of horizontal axis in the Figure 6 was revised.

Figure 6. Measured and calculated tensile (a) modulus and (b) strength of PA6 composites.
